# Burnout among Chinese live streamers: Prevalence and correlates

Shi Chen[1,2,3], Hanqin Wang[1,3], Shang Yang[1,3], Fushen Zhang[1,3], Xiao Gao[1,3], Ziwei Liu [1,3]*

1 Key Laboratory of Molecular Epidemiology of Hunan Province, Hunan Normal University, Changsha, China, 2 National Clinical Research Center for Mental Disorders, The Second Xiangya Hospital of Central South University, Changsha, China, 3 School of medicine, Hunan Normal University, Changsha, 410013, China

* liuziwei@hunnu.edu.cn

**Data Availability Statement:** All relevant data are within the manuscript and its Supporting Information files.

**Funding:** This work was supported by the Hunan Normal University undergraduates innovative

## Abstract

### Background

The prevalence of burnout among live streamers remains largely unknown. This study aims to investigate the prevalence and factors associated with burnout among Chinese live streamers.

### Methods

A cross-sectional study recruited 343 full-time live streamers from 3 companies in Changsha city. Socio-demographic and occupational characteristics were collected using self-designed items. Job stress was assessed using the Job Content Questionnaire (JCQ-22), while supervisor and coworker support were evaluated using the last 8 items of the JCQ-22. Burnout was assessed using the 17-item Chinese version of the Maslach Burnout Inventory-Human Services Survey (MBI-HSS).

### Results

Our findings revealed that 30.6% of live streamers experienced burnout. Lower levels of education (OR = 2.65 and 3.37, $p = 0,005$ and 0.003), higher monthly income (OR = 10.56 and 11.25, both $p = 0.003$), being an entertainment-oriented streamer (OR = 2.13, $p = 0.028$), continuous walking during live streams (OR = 2.81, $p = 0.006$), significant drop in follower count (OR = 2.65, $P = 0.006$), live streaming during the daytime (OR = 3.75, $p = 0.001$), and higher support from supervisors and coworkers (OR = 3.66, $p = 0.001$) were positively associated with burnout. However, the effects of education and drop in followers on burnout were not significant in the multivariate logistic models ($p = 0.321$ and 0.988).

### Conclusions

Burnout among Chinese live streamers is associated with income, being an entertainment streamer, engaging in continuous walking during live streams, conducting live streams during the daytime, and experiencing excessive support from supervisors and coworkers.

experiment project and entrepreneurship program (2019116), Hunan Normal University major event social stability risk assessment center 2021-year program (2021WP08). The funders had no role in study design, decision to publish, or preparation of the manuscript.

**Competing interests:** The authors have declared that no competing interests exist.

## Background

Live streamer is defined as an individual who broadcasts live content over the internet. The livestreaming industry experienced rapid growth and holds significant economic value. According to the 2020-year report of China Internet Network Information Center, the total amount of financing and the market size in the livestreaming industry reached 6.23 billion and 193.03 billion, respectively [1]. This industry generated a substantial number of job opportunities. Live streaming, as a high-income work, has attracted a large influx of individuals. A survey conducted in China on the income of live streamers showed that 93% of streamers earn a monthly income exceeding 4500 Chinese Yuan (CNY), equal to 706 dollars, which is significantly higher than the national per capita income of 2682 CNY [2]. Furthermore, this work has a low barrier to entry. Advancements in internet communication technology, smartphone, applications, and the reduction of associated cost have provided ordinary individuals with the necessary hardware and equipment to become live streamers. In addition, becoming a streamer requires minimal special skill or training. Being a live streamer has emerged as an appealing and popular career choice. As of the end of 2020, the number of live streamer accounts in China had reached 130 million, with over 43 thousand new accounts being added every day [1].

Live streaming is a relatively new career characterized by an underdeveloped occupational safety and health administration compared to established operational systems. Live streamers often face excessive working hours, heavy workload, and even disruptions to their normal pace of life. Additionally, they encounter many challenges during work such as maintaining a fixed posture for a long period of time, experiencing rapid fluctuation in fans count, various types of cyberbullying, and lacking of scientific and reasonable evaluation methods and systems. These reasons add significant work pressure or job-related stress. Prolonged exposure to excessive job stress can result in a range of physical and mental health problems. However, the majority of live streamers have not received sufficient occupational health support beyond only basic medical insurance. The remaining individuals do not even have any insurance coverage purchased by their companies. The combination of substantial job stress, job-related health problems, and the absence of labor and social security measures contributed to a high turnover rate, overall employee dissatisfaction, and increasing occupational hazard.

Burnout was defined as a state of physical and mental exhaustion experienced by workers in the service industry as a result of prolonged, high-intensity, and heavy workload [3]. It is characterized by emotional exhaustion, cynicism, and negative self-evaluation [4]. Burnout is a significant consequence of chronic, unresolved job stress, which not only diminishes work efficiency and quality but also poses risks to physical and mental illness, such as depression and insomnia [5,6]. Implementing interventions can effectively prevent occupational injuries associated with burnout. While extensive researches had investigated the prevalence and correlates of burnout among various populations, such as physician, nurse, and teachers, there is lack of prior surveys specifically targeting live streamers [7–9].

This study aimed to examine the cross-sectional status and correlates of burnout among Chinese live streamers. The objective was to identify the key risk factors associated with burnout, provide novel insights for future research of intervention mechanisms, and promote the occupational health of practitioners.

## Methods

### Participants

This cross-sectional survey recruited 369 participants from 3 broadcasting companies located in Changsha city, Hunan province. These companies provided livestreaming service to the

public through the internet. The inclusion criteria for study participants were as follows: (1) being a full-time live streamer; (2) over 16 years old; and (3) having signed a contract with a broadcasting company. Participants would be excluded if they: (1) had not signed a contract with a broadcasting company; (2) were younger than 16; (3) had difficulties in reading or communicating; and (4) were unable to independently complete questionnaires using electronic devices.

## Procedure

This study was reviewed and approved by the Medical Ethics Committee of Hunan Normal University (permit number: 2021–283). The study was conducted from October 2021 to December 2021 in 3 broadcasting companies, which offered various types of full-time jobs for livestreaming. Prior to conducting the study, permission was obtained from all participating companies. Trained investigators, who were post-graduates from the school of medicine of Hunan Normal University, conducted the recruitment. The investigators explained the purpose of the study to potential participants and directed interested individuals to provide informed consent. Participants completed the assessment through electronic survey platforms (WWW.WJX.CN and WENJUAN.COM). In the case of participants aged 16 to 18 years, informed consent was obtained from their guardians. A total of 369 participants provided electronic informed consent and completed questionnaire.

## Measures

Socio-demographic factors. Socio-demographic factors assessed in the current study included age, gender, education and income. Age was calculated based on the participant's birthdate. We categorized the original numeric data on age into 2 groups: 16 to 25 years and more than 25 years. Education was classified into four categories: junior high school and below, high/vocational high/technical secondary school, junior college, and undergraduate and above. Income was originally collected as self-reported personal monthly income. For this study, we categorized income into three groups: 10,000 CNY and above, 10,000 to 5,000 CNY, 5,000 to 1,500 CNY. A monthly income of 1,540 yuan represented the minimum income in Changsha city in 2021 [10]. Therefore, self-reported incomes below 1,500 yuan were considered false reports and treated as missing data.

**Occupational factors.** Occupational factors included work tenure, type of live streamer, work posture, fluctuation in followers count, rest days per month, length of livestreaming in a 24-hours day, livestreaming time slot, work performance, and experienced cyberbullying or not. Work tenure was defined as the duration from the participant's initial hire date at the current company to the date of our interview. Hire dates were collected and utilized to calculate work tenure. We categorized work tenure into 2 groups: less than 1 year and more than 1 year. Type of live streamer was assessed by the question of "The type of live streamer you belong to ___?" with 3 options including entertainment streamer, game streamer, and other. Entertainment streamer included singing, chatting, dance, playing musical instruments, serving as an emcee, and net jockey. Game streamer included individuals who streamed various types of games. Other streamers included those streamed teaching, food-related content, outdoor activities, language instruction, financial topics, fitness, investment, and other types not mentioned. Work posture refers to the position and alignment of the body while performing work tasks. Work posture was assessed by asking participants the question, "What is your work posture?" They could choose from 4 options: continuous standing, continuous sitting, continuous walking, and periodic posture change. Fluctuation in followers count was defined as the change in the number of followers over the past month, and was assessed by asking, "In the past month,

has there been any change in your number of fans?" The response options included significantly increased, increasing slowly, no significant change, and significantly decreased. Rest days per month were assessed by asking, "How many rest days do you have every month?". The original numeric data were categorized into 3 groups: more than 4 days, 4 days, and less than 4 days. The length of live stream in a 24-hours (24h) day was assessed by asking, "How long do you livestream in a 24-hours day"The response options included 2-4hours, 4-6hours, 6-8hours, and >8hours. The livestreaming time slot was assessed by asking, "What is your livestream time slot?" The multiple-choice options initially included 8:00–12:00, 12:00–14:00, 14:00–19:00, 19:00–24:00, and 0:00–8:00. However, for the purposes of analysis, it was simplified to daytime (8:00–19:00) and evening (19:00–8:00). Work performance was assessed by asking, "In the past 6 months, have you completed the assigned livestreaming task every month?" The response options initially included never completed on time, completed 1–2 months, completed 3–4 months, and completed 5–6 months. However, for clarity and consistency, these options were modified to represent the following ratings: bad (never completed on time), general (completed 1–2 months), good (completed 3–4 months), and very good (completed 5–6 months). Cyberbullying was assessed by asking participants, "Have you experienced cyberbullying?" the response options were simplified to yes or no.

**Job stress.**   Job stress was defined as the imbalance between an individual's abilities and the demand of their work [11]. Job stress was assessed using the Job Content Questionnaire (JCQ-22), consisting of 22 items rated on a 4-point score from 1 to 4. Out of the 22 items, 14 items were used to quantify the job stress experienced by each participant. Among these 14 items, the sum score of 5 items represented the job demands, while the score of 9 items represented the job control. The internal consistency of 14 items in this study showed good reliability, with a Cronbach's alpha of 0.95. Drawing upon the job demand-control model theory, job stress was represented by calculating the ratio of job demands and control (D/C ratio), weighted by the number of items. The formula used as follows: D/C ratio = (job demand score / job control score) *(9/5). A ratio > 1 indicated high job stress, while a ratio < 1 indicated low job stress [12].

**Supervisor and coworker support.**   Supervisor and coworker support was assessed using the last 8 items of JCQ-22 [13]. In the current study, the Cronbach's alpha was 0.94. A total score ranging from 0 to 32 was used, with a higher score indicating high support. No specific cutoff value was identified; Therefore, tertiles were used to classify the different levels of support. In our sample, the top tertile corresponded to a score of 24 or above, while the bottom tertile corresponded to a score of 18 or below. We classified supervisor and coworker support as high (score of 25 or above), moderate (score between 24 and 18), and low (score of 17 or less).

**Burnout.**   Burnout was assessed using the revised Chinese version of the Maslach burnout inventory-human services survey (MBI-HSS) [14]. Each item was rated on a 6-point score, ranging from 0 (none of the time) to 6 (every day). MBI-HSS comprises three dimensions: a 7-item subscale of emotional exhaustion, a 3-item subscale of depersonalization, and a 7-item subscale of personal accomplishment. Burnout was defined as a combination of a high score on emotional exhaustion and a high score on depersonalization. High scores in either of the dimensions were defined as scores exceeding the top tentile. In the current study, the Cronbach's alpha for the MBI-HSS was 0.96. The top tertile of emotional exhaustion corresponded to a score of 27, while a score of 10 represented the threshold for depersonalization. Therefore, individuals who scored >27 on the emotional exhaustion and scored > 10 on the depersonalization met the criteria for burnout.

## Statistical analysis

All statistical analyses were performed using SPSS software version 22.0. Mean and standard deviation displayed for continuous variables. Frequency and percentages were displayed for

categorical variables. *t*-test or Chi-square test were used to compare socio-demographic and occupational differences between burnout and non-burnout groups. Kendall's tau-b/c were used to assess the inter-correlations between variables. Univariate and multivariate logistic regression analysis was conducted, with socio-demographics and occupational characteristics as the independent variables, while burnout was the dependent variable. Odds ratio and 95% confidence intervals (CI) were reported, with *p*-values of 0.05 considered statistically significant.

## Results

### Participant characteristics

Among 369 eligible participants whom we approached, 11 were excluded due to missing values on variables, and 15 were excluded due to logical errors in their questionnaire responses. Finally, 343 participants were included in the current study resulting in a quality rate of 92.95%.

The socio-demographic characteristics of the 343 participants were presented in the upper half of Table 1. The majority of streamers were female (56.9%), aged 16 to 24 years (56.0%), and had a monthly income between 5,000 and 10,000 yuan (48.1%). Additionally, 37.9% of participants had a junior college educational background.

### The proportion of burnout and occupational characteristics

The lower half of Table 1 presents the occupational characteristics of the sample. Among the 343 live streamers, 30.6% reached the level of burnout. The majority of streamers had more than 1 year of work tenure (70.6%), worked in the entertainment live streaming (38.3%), maintained a continuous sitting work posture (49.3%), experienced slow growth in their followers count (42.0%), had more than 4 days of rest per month (42.7%), conducted live streams of 4–6 hours in length (34.4%), scheduled their live streams during the daytime (72.8%), demonstrated good work performance (35.3%), experienced cyberbullying (59.5%), and reported a high level of job stress (67.8%). Moreover, the level of supervisor and coworker support was rated as medium (87.5%).

### Comparisons between burnout and non-burnout group

Columns 3 to 5 of Table 1 display the comparisons between the burnout and non-burnout groups regarding socio-demographic and occupational characteristics. Among all socio-demographic factors, significant differences were observed in education (19.4% vs. 30.0% vs. 39.0% vs. 44.7%, *p* = 0.008) and income (4.9% vs. 44.8% vs. 44.7%, *p* < 0.001). Regarding occupational factors, significant differences were found in work posture (*p* < 0.001), fluctuation in followers count (*p* = 0.012), live stream time slot (*p* < 0.001), and supervisor and coworker support (*p* < 0.001). These findings indicated that lower education levels, higher income, poor work posture, a drop in followers count, daytime live streaming, and supervisor and coworker support may be correlated with burnout. Detailed comparisons are provided in Table 1.

### Correlates of burnout

Table 2 presents the results of univariate and multivariate logistic regressions that examine the socio-demographic and occupational factors associated with burnout. The univariate analysis revealed that burnout was correlated with education (OR = 1.78–3.37, *p* = 0.003–0.070), income (OR = 15.75 and 15.83, all *p* < 0.001), type of live streamer (OR = 1.95, *p* = 0.030),

**Table 1. Characteristics of live streamers between burnout vs. non-burnout groups.**

| Variables | Total | Burnout | | p |
| --- | --- | --- | --- | --- |
| | | No | Yes | |
| n | 343 (100.0) | 238 (69.4) | 105 (30.6) | |
| **Socio-demographic factors** | | | | |
| Gender | | | | |
| Male | 148 (43.1) | 109 (73.6) | 39 (26.4) | 0.160 |
| Female | 195 (56.9) | 129 (66.2) | 66 (33.8) | |
| Age | | | | |
| 16 ~ 24 | 192 (56.0) | 133 (69.3) | 59 (30.7) | 0.958 |
| 25 ~ 53 | 151 (44.0) | 105 (69.5) | 46 (30.5) | |
| Educational background | | | | |
| Undergraduate and above | 98 (28.6) | 79 (80.6) | 19 (19.4) | |
| Junior college | 130 (37.9) | 91 (70.0) | 39 (30.0) | |
| High, vocational high, or technical secondary school | 77 (22.4) | 47 (61.0) | 30 (39.0) | |
| Junior high school and below | 38 (11.1) | 21 (55.3) | 17 (44.7) | **0.008** |
| Monthly income (CNY) | | | | |
| > 10000 | 94 (36.2) | 52 (55.3) | 42 (44.7) | |
| 5000 ~ 10000 | 125 (48.1) | 69 (55.2) | 56 (44.8) | |
| 1500 ~ 5000 | 41 (15.8) | 39 (95.1) | 2 (4.9) | **<0.001** |
| Missing | 83 | | | |
| **Occupational factors** | | | | |
| Work tenure | | | | |
| < 1 year | 101 (29.4) | 75 (31.5) | 26 (25.7) | 0.206 |
| > = 1 year | 242 (70.6) | 163 (67.4) | 79 (32.6) | |
| Type of live streamer | | | | |
| Entertainment streamer | 131 (38.3) | 84 (64.1) | 47 (35.9) | |
| Game streamer | 118 (34.4) | 81(68.6) | 37 (31.4) | |
| Other | 94 (27.4) | 73 (77.7) | 21 (22.3) | 0.092 |
| Work posture | | | | |
| Continuous walking | 58 (16.9) | 26 (44.8) | 32 (55.2) | |
| Continuous standing | 45 (13.1) | 29 (64.4) | 16 (35.6) | |
| Continuous sitting | 169 (49.3) | 137 (81.1) | 32 (18.9) | |
| Periodic posture change | 71 (20.7) | 46 (64.8) | 25 (35.2) | **<0.001** |
| Fluctuation of followers count | | | | |
| Increased significantly | 51 (14.9) | 34 (66.7) | 17 (33.3) | |
| Increased slowly | 144 (42.0) | 107 (74.3) | 37 (25.7) | |
| No significant change | 84 (24.5) | 63 (75.0) | 21 (25.0) | |
| Decreased significantly | 64 (18.7) | 34 (53.1) | 30 (46.9) | **0.012** |
| Rest days per month | | | | |
| < 4 days | 112 (33.0) | 75 (67.0) | 37 (33.0) | |
| 4 days | 85 (25.2) | 59 (69.4) | 26 (30.6) | |
| > 4 days | 142 (42.7) | 100 (70.4) | 42 (29.6) | 0.836 |
| Length of live stream in a 24h day | | | | |
| 2~4 h | 78 (22.7) | 55 (70.5) | 23 (29.5) | |
| 4~6 h | 118 (34.4) | 85 (72.0) | 33 (29.0) | |
| 6~8 h | 109 (31.8) | 76 (69.7) | 33 (30.3) | |
| >8 h | 38 (11.1) | 22 (57.9) | 16 (42.1) | 0.423 |
| Live stream time slot | | | | |

*(Continued)*

**Table 1.** (Continued)

| Variables | Total | No | Yes | p |
|---|---|---|---|---|
| | | | Burnout | |
| Evening (19:00 ~ 8:00) | 93 (27.2) | 84 (90.3) | 9 (9.7) | |
| Daytime (8:00 ~19:00) | 249 (72.8) | 153 (61.4) | 96 (38.6) | **<0.001** |
| Work performance | | | | |
| Bad | 44 (12.8) | 33 (75.0) | 11 (25.0) | |
| General | 78 (22.7) | 57 (73.1) | 21 (26.9) | |
| Good | 121 (35.3) | 82 (67.8) | 39 (32.2) | |
| Very good | 100 (29.2) | 66 (66.0) | 34 (34.0) | 0.606 |
| Cyberbullying | | | | |
| No | 139 (40.5) | 99 (71.2) | 40 (28.8) | |
| Yes | 204 (59.5) | 138 (68.1) | 65 (31.9) | 0.543 |
| Job stress | | | | |
| High | 236 (68.8) | 160 (67.8) | 76 (32.2) | |
| Low | 107 (31.2) | 78 (72.9) | 29 (27.1) | 0.342 |
| Supervisor and coworker support | | | | |
| High | 133 (38.8) | 67 (50.4) | 66 (49.6) | |
| Medium | 104 (30.3) | 91 (87.5) | 13 (12.5) | |
| Low | 106 (30.9) | 80 (75.5) | 26 (24.5) | **<0.001** |

work posture (OR = 2.33–5.27, $p$ = <0.001–0.020), fluctuation in follower count (OR = 2.65, $p$ = 0.006), live stream time slot (OR = 5.86, $p$ < 0.001), as well as supervisor and coworker support (OR = 2.28 and 6.89, $p$ <0.001 and 0.027).

Considering the large number of missing values for the income variable, multivariate analyses were conducted using two models: model 1, which excluded income, with a sample size of 343, and model 2, which included income, with a sample size of 259. In model 1, burnout was found to be associated with being an entertainment streamer (OR = 2.13, 95% CI: 1.08–4.20), continuous walking as a work posture (OR = 2.81, 95%CI: 1.35–5.85), live streaming during the daytime (OR = 3.75, 95%CI: 1.69–8.30), and a high level of supervisor and coworker support (OR = 3.66, 95%CI: 1.76–7.64). In model 2, the results showed that higher income (OR = 10.56 and 11.25, 95%CI: 2.18–51.08 and 2.29–55.10, respectively), continuous walking as a work posture (OR = 2.28, 95%CI: 1.02–5.08), live streaming during the daytime (OR = 3.91, 95%CI: 1.43–10.67), and a high level of supervisor and coworker support (OR = 3.90, 95%CI: 1.58–9.61) were associated with burnout. These findings indicated that being an entertainment streamer, engaging in continuous walking during live streams, conducting live streams during the daytime, and experiencing excessive support from supervisors and coworkers are independent predictors of job burnout among live streamers.

### The inter-correlations of socio-demographic and occupational characteristics

It is noteworthy that the statistical significance of "education" and "fluctuation in followers count" observed in the univariate analyses disappeared in the multivariate analyses. Kendall's tau-b/c was employed to examine the inter-correlations among the study variables. Table 3 presents the correlation coefficients between the variables. The results revealed that education was significantly correlated with work posture ($r$ = 0.174, $p$ < 0.001), fluctuation in followers count ($r$ = 0.105, $p$ = 0.023), and the supervisor and coworker support ($r$ = 0.201, $p$ < 0.001).

**Table 2. Logistic regression analyses of demographic and occupational characteristics relating to burnout.**

| | Univariate analysis | | Multivariate Model 1 | | Model 2 | |
|---|---|---|---|---|---|---|
| Variables | OR (95% CI) | *p* | OR (95% CI) | *p* | OR (95% CI) | *p* |
| *n* included in analysis | 343 | | 343 | | 259 | |
| **Socio-demographic factors** | | | | | | |
| Education (undergraduate and above as reference) | | | | | | |
| Junior college | **1.78 (0.95, 3.33)** | **0.070** | 1.12 (0.55, 2.30) | 0.748 | 0.89 (0.39, 2/03) | 0.790 |
| High, vocational high, or technical secondary school | **2.65 (1.35, 5.23)** | **0.005** | 1.53 (0.70, 3.34) | 0.286 | 1.63 (0.67, 3.94) | 0.274 |
| Junior high school and below | **3.37 (1.49, 7.58)** | **0.003** | 1.51 (0.61, 3.74) | 0.365 | 1.36 (0.49, 3.80) | 0.551 |
| Monthly income (1500 ~ 5000 as reference) | | | | | | |
| 5000 ~ 10000 | **15.83 (3.66, 68.42)** | **<0.001** | | | **10.56 (2.18, 51.08)** | **0.003** |
| > 10000 | **15.75 (3.59, 69.05)** | **<0.001** | | | **11.25 (2.29, 55.10)** | **0.003** |
| **Occupational factors** | | | | | | |
| Type of live streamer (other as reference) | | | | | | |
| Entertainment streamer | **1.95 (1.07, 3.55)** | **0.03** | **2.13 (1.08, 4.20)** | **0.028** | 2.08 (0.97, 4.43) | 0.058 |
| Game streamer | 1.59 (0.85, 2.96) | 0.145 | 1.47 (0.73, 2.97) | 0.276 | 1.35 (0.62, 2.92) | 0.444 |
| Work posture (continuous sitting as reference) | | | | | | |
| Periodic posture change | **2.33 (1.25, 4.33)** | **0.008** | 1.91 (0.95, 3.84) | 0.067 | 1.58 (0.72, 3.47) | 0.251 |
| Continuous standing | **2.36 (1.15, 4.86)** | **0.02** | 1.44 (0.64, 3.20) | 0.373 | 1.25 (0.50, 3.10) | 0.623 |
| Continuous walking | **5.27 (2.76, 10.04)** | **<0.001** | **2.81 (1.35, 5.85)** | **0.006** | **2.28 (1.02, 5.08)** | **0.043** |
| Fluctuation of followers count (no significant change as reference) | | | | | | |
| Increased slowly | 1.04 (0.56, 1.93) | 0.908 | 0.82 (0.40, 1.66) | 0.591 | 0.55 (0.23, 1.30) | 0.177 |
| Increased significantly | 1.50 (0.70, 3.22) | 0.298 | 0.99 (0.41, 2.37) | 0.988 | 0.65 (0.23, 1.82) | 0.417 |
| Decreased significantly | **2.65 (1.32, 5.31)** | **0.006** | 1.57 (0.67, 3.38) | 0.321 | 0.88 (0.34, 2.27) | 0.796 |
| Live stream time slot (night as reference) | | | | | | |
| Daytime | **5.86 (2.81, 12.19)** | **<0.001** | **3.75 (1.69, 8.30)** | **0.001** | **3.91 (1.43, 10.67)** | **0.008** |
| Supervisor and coworker support (medium as reference) | | | | | | |
| Low | **2.28 (1.10, 4.72)** | **0.027** | 1.48 (0.66, 3.33) | 0.337 | 2.15 (0.80, 5.75) | 0.127 |
| High | **6.89 (3.52, 13.52)** | **<0.001** | **3.66 (1.76, 7.64)** | **0.001** | **3.90 (1.58, 9.61)** | **0.003** |

Additionally, the fluctuation in followers count correlated with income ($r = 0.121$, $p = 0.024$), and supervisor and coworker support ($r = 0.170$, $p < 0.001$). We speculate that a low level of education and fluctuation in followers count, particularly a drop in followers, may not be independent risk factors for burnout. The impact of education and a drop in followers count might be influenced or masked by other strong predictors.

**Table 3. Inter-correlations between demographic and occupational characteristics.**

| Variables | 1 | 2 | 3 | 4 | 5 |
|---|---|---|---|---|---|
| 1. Education | | | | | |
| 2. Work posture | 0.174*** | | | | |
| 3. Fluctuation in follower count | 0.105* | 0.202*** | | | |
| 4. Income | -0.010 | 0.015 | 0.121* | | |
| 5. Support from supervisors and coworkers | 0.201*** | 0.242*** | 0.170*** | 0.220*** | |
| 6. Type of live streamer | 0.072 | -0.013 | 0.054 | 0.008 | 0.009 |

*indicates *p*<0.05

*** indicates *p*<0.001.

## Discussion

### Summary of findings

The present study found that 30.6% of live streamers experienced burnout. Burnout was found to be correlated with education, income, type of streamer, work posture, fluctuation in followers count, live stream time slot, and supervisor and coworker support. Specifically, lower educational attainment and higher monthly income showed a positive correlation with burnout. Regarding occupational factors, being an entertainment streamer, engaging in continuous walking during live streams, experiencing a significant drop in follower count, conducting live streams during the daytime, and receiving higher support from supervisors and coworkers were positively correlated with burnout. However, the impact of education and drop in followers on burnout may be influenced or masked by other strong correlates.

### The proportion of burnout

Our findings indicated that 30.6% of live streamers experienced burnout. Comparing our results with similar studies is challenging due to limited existing research among the live streamer population. However, our findings are consistent with expectations and suggest a relatively high proportion of burnout among live streamers. Burnout has been extensively documented among various professions, including physicians, nurses, teachers, and police officers. For instance, Rotenstein et al. conducted a systematic review on physician burnout and reported an overall proportion of 67% [7]. The global proportion of burnout among nurses is reported to be 11.23% [15]. Among teachers, the proportion of burnout rates range from 25.12% to 74% [16]. Studies on police officers indicate a proportion ranging from 28% to 32% [17]. Therefore, the burnout among live streamers appears to be lower than that among physicians but higher than that among nurses, comparable to teachers and police officers.

It is important to note that differences in work content and forms of labor between different professions, as well as variations in measurement tools and classification criteria, contribute to these disparities. In our study, we utilized a revised Chinese version of the 17-item Maslach Burnout Inventory specifically tailored to the Chinese population, instead of the original 22-item version. However, there is currently no universally accepted cutoff score for this scale. We employed the top tertile as the cutoff, a method used in other study as well [18]. The top tertile scores for emotional exhaustion and cynicism were 28 and 11, respectively, which are comparable to commonly used cutoffs (27 and 11) [19,20]. While this may appear coincidental, it is crucial to consider that the decreased total score and the corresponding increase in the cutoff resulted in fewer individuals being classified as experiencing burnout. Therefore, the reported proportion of 30.6% may underestimate the actual occurrence status. Nevertheless, this does not diminish the conclusion that burnout is prevalent among the live streamer.

### Correlates of burnout

The present study identified several factors associated with burnout among live streamers, including being an entertainment streamer, engaging in continuous walking during live streams, live streaming during the daytime, higher income, excessive support from supervisors and coworkers, lower education level, and experiencing a drop in followers. Firstly, we speculate that the increased risk of burnout among entertainment streamers and streamers who engage in continuous walking might be attributed to the higher physical activity intensity involved. Game streamer included individuals who streamed various types of games. Other steamers included

those streamed teaching, food-related content, outdoor activities, language instruction, financial topics, fitness, investment, and other types not mentioned. Entertainment streamers included singing, chatting, dance, playing musical instruments, serving as an emcee, and net jockey. Entertainment streamers seem to have higher levels of physical activity compared to the other two types of streamers. Similarly, continuous walking during live streams leads to higher physical exertion and increased intensity of movement compared to sitting continuously, although the study did not measure the physical labor workload among streamers. This finding aligns with research conducted on physicians and teachers, where similar patterns were observed [21].

Secondly, burnout associated with live streaming during the daytime is likely because the prime time for live streaming is usually in the evening when the general population has time to watch. Therefore, for high-performing, popular, and influential streamers, their work is primarily scheduled for the evening. In contrast, streamers with lower performance and influence may be assigned to stream during the daytime. This could be related to their states of burnout, but the specific mechanisms need further investigation.

Thirdly, contrary to findings in other populations, higher income and excessive support from supervisors and coworkers were not protective factors against burnout in this study; instead, they were risk factors [22,23]. It is speculated that in this study, income and social support are not the causes of burnout but rather co-variable. Generally, high-performing streamers who generate more economic benefits tend to have higher income and receive more support and attention from colleagues. However, they also face higher work intensity, which increases the risk of burnout. However, this hypothesis requires further research for confirmation.

Fourthly, lower education level and a drop in followers increase the risk of burnout, although these effects may be confounded by other factors. Studies in other populations have shown that education level is associated with burnout, with a higher risk of burnout among individuals with lower education [24]. As education precedes burnout in time, the relationship between the two is clear in most populations. Therefore, low education level is likely one of the etiological factors for burnout. It could be due to a lack of knowledge and coping strategies for dealing with burnout. Strengthening monitoring, occupational protection, and training for streamers with lower education levels may be one of the solutions.

Finally, our understanding of the prevalence and factors contributing to burnout among live streamers remains limited. Our findings have implications for the future establishment of occupational protection for streamers. It is not only important to provide a comprehensive description of the characteristics of streamers' work but also to establish the causal chain leading to burnout. To document the key association and modifiable factors contributing to burnout will allow us to propose intervention measures and methods to establish a comprehensive labor protection system for streamers' occupation.

## Limitations

This study had several limitations. Firstly, although our sample was recruited from three companies in the urban city of Changsha, the generalizability of our findings to other regions may be limited. Additionally, our study only included full-time streamers who had signed contracts with companies, which may restrict the generalizability of our findings to part-time or self-employed streamers. Secondly, we did not assess a broader range of occupational factors that could potentially impact burnout, such as workload, work intensity, and fatigue. Future research should consider including these and other relevant factors as potential correlates of burnout. Thirdly, due to limitations in our measurement tools and assessment criteria, measurement errors that are difficult to avoid may have been introduced. Further research should employ more valid instruments to assess burnout among streamers.

## Conclusion

In this study involving Chinese full-time live streamers, we observed a relatively high prevalence of burnout. Various factors were found to be correlated with burnout, including education level, income, being an entertainment streamer, engaging in continuous walking during live streams, experiencing a significant drop in follower count, conducting live streams during the daytime, and experiencing excessive support from supervisors and coworkers. Among these correlates, the impact of low education level and a drop in followers on burnout may be influenced or overshadowed by other strong factors. These findings have implications for further theoretical development and research on the underlying mechanisms of burnout in the live streaming industry. Additionally, interventions should address both individual traits and work environment conditions to promote positive and long-lasting occupational health outcomes.

## Supporting information

**S1 Raw data.**
(XLSX)

## Acknowledgments

The authors would like to acknowledge the staff from Wan Hui media co., LTD of Changsha.

## Author Contributions

**Conceptualization:** Shi Chen, Ziwei Liu.

**Funding acquisition:** Shi Chen, Ziwei Liu.

**Investigation:** Hanqin Wang.

**Methodology:** Shang Yang.

**Project administration:** Fushen Zhang.

**Supervision:** Fushen Zhang.

**Visualization:** Xiao Gao.

**Writing – original draft:** Shi Chen.

**Writing – review & editing:** Ziwei Liu.

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
