## [Decision Letter · Decision Letter 0]

17 Aug 2023

PONE-D-23-18174Burnout prevalence and its associated factors among Chinese webcastersPLOS ONE

Dear Dr. Liu,

Thank you for submitting your manuscript to PLOS ONE. After careful consideration, we feel that it has merit but does not fully meet PLOS ONE’s publication criteria as it currently stands. Therefore, we invite you to submit a revised version of the manuscript that addresses the points raised during the review process.

Dear authors,

We have thoroughly reviewed your paper and find it to be of significant interest. However, before we can consider accepting it, there are some major revisions that need to be addressed. Kindly respond to all the comments provided by the reviewers. Additionally, we suggest that the paper undergo proofreading by a native English speaker to ensure clarity and correctness.

Furthermore, we recommend the removal of the section in the discussion part. Instead, we encourage you to delve into various issues within this section, engaging with professionals from different fields such as healthcare providers and teachers.

Finally, we kindly request that you include broader implications of your study for the readers. This will enhance the overall significance and relevance of your research. Thank you for your efforts, and we look forward to reviewing your revised paper.

Best regards, 

Othman A. Alfuqaha

We look forward to receiving your revised manuscript.

Kind regards,

Othman A. Alfuqaha, Ph.D.

Academic Editor

PLOS ONE

“This work was supported by the Hunan Normal University undergraduates innovative experiment project and entrepreneurship program (2019116), Hunan Normal University major event social stability risk assessment center 2021-year program (2021WP08).”

Additional Editor Comments:

Dear authors,

We have thoroughly reviewed your paper and find it to be of significant interest. However, before we can consider accepting it, there are some major revisions that need to be addressed. Kindly respond to all the comments provided by the reviewers. Additionally, we suggest that the paper undergo proofreading by a native English speaker to ensure clarity and correctness.

Furthermore, we recommend the removal of the section in the discussion part. Instead, we encourage you to delve into various issues within this section, engaging with professionals from different fields such as healthcare providers and teachers.

Finally, we kindly request that you include broader implications of your study for the readers. This will enhance the overall significance and relevance of your research. Thank you for your efforts, and we look forward to reviewing your revised paper.

Please do not forget the attachment PDF to answer the reviewer number 2.

Best regards,

Reviewers' comments:

Reviewer's Responses to Questions

**Comments to the Author**

1. Is the manuscript technically sound, and do the data support the conclusions?

Reviewer #1: Yes

Reviewer #2: Yes

2. Has the statistical analysis been performed appropriately and rigorously? 

Reviewer #1: Yes

Reviewer #2: Yes

3. Have the authors made all data underlying the findings in their manuscript fully available?

Reviewer #1: No

Reviewer #2: No

4. Is the manuscript presented in an intelligible fashion and written in standard English?

Reviewer #1: No

Reviewer #2: No

5. Review Comments to the Author

Reviewer #1: The paper addresses an important and timely topic since job burnout is pervasive and new risks are emerging in web-based activities.

I do have a few comments, however, on which I would like to ask the authors to respond.

(1) Dependent variable: The classification into groups with high or low burnout scores should be explained in more detail.

- Does "mean of each dimension" refer to the sample mean? And were no external norms used? If so, approximately half of the sample is considered to be at risk, even if the values are, for example, in the lower range compared to other professions. That would make it difficult to determine the actual risk of burnout.

- “Further, both high-EE and high-DP (above average) were judged as the group with high-burnout”: does this mean that PA was not considered for the classification?

- Studies were cited which used a similar approach. However, the MBI provides norm-based cutoff values for "low risk," "moderate risk," and "high risk." Why were these not considered?

(2) Discussion, Main findings: It is useful to briefly repeat the main results, but the statistical metrics (CI, OR) should not be repeated.

(3) Discussion, Dynamic risk factors:

- I didn't entirely understand why a nighttime broadcast is less dangerous than a daytime one. Is night work the norm in this profession?

- Overall, the discussion could make more reference to theory and previous studies.

(4) Discussion, Protective factor: “Interpersonal support is unlikely to directly cause burnout”: However, the analysis model tested direct influence (and it was significant), right? No moderator effects of "interpersonal support" were tested either (if I interpret it correctly). Thus, these conclusions can actually only represent hypotheses for future research.

(5) Minor comments:

- The p-value is usually written in lower case.

- I am not a native English speaker, but a few phrases seemed to me linguistically not quite correct. I recommend having the text checked again.

Reviewer #2: The results were not adequately discussed with very scant reference to existing literature.

This manuscript would most certainly require the services of an English language editor for major revisions.

Other comments have been included in the manuscript which has been attached.

6. PLOS authors have the option to publish the peer review history of their article (what does this mean?). If published, this will include your full peer review and any attached files.

Reviewer #1: **Yes: **Verena Hofmann

Reviewer #2: No

---

## [Author Response · Author response to Decision Letter 0]

20 Feb 2024

Response to Reviewers

Dear All,

Thank you very much for your thorough review of our manuscript. During the revision process, we identified some statistical errors, prompting us to reexamine the raw data and conduct all the statistical analyses anew. The updated statistical results significantly differ from the previous ones, leading us to revise the logic and content of the paper accordingly. As a result, it took us longer than anticipated. We kindly request you to review the revised manuscript once again, and we appreciate your efforts and understanding.

Ziwei Liu

Feb.15 2024

Reviewer #1: The paper addresses an important and timely topic since job burnout is pervasive and new risks are emerging in web-based activities.

I do have a few comments, however, on which I would like to ask the authors to respond.

(1) Dependent variable: The classification into groups with high or low burnout scores should be explained in more detail.

- Does "mean of each dimension" refer to the sample mean? And were no external norms used? If so, approximately half of the sample is considered to be at risk, even if the values are, for example, in the lower range compared to other professions. That would make it difficult to determine the actual risk of burnout.

Response：Through literature review, we found that the practice of simply using the total score of the Maslach Burnout Inventory-Human Services Survey (MBI-HSS) to indicate burnout does not adhere to the original scale's guidelines. Therefore, we have redefined the categorization of burnout based on the following two studies. 

[7] Rotenstein LS, Torre M, Ramos MA, Rosales RC, Guille C, Sen S, Mata DA. Prevalence of Burnout Among Physicians: A Systematic Review. JAMA. 2018;320(11):1131-1150. doi: 10.1001/jama.2018.12777. 

[16] Upton D, Mason V, Doran B, Solowiej K, Shiralkar U, Shiralkar S. The experience of burnout across different surgical specialties in the United Kingdom: a cross-sectional survey. Surgery. 2012;151(4):493-501. doi: 10.1016/j.surg.2011.09.035. 

In the revised manuscript, we provided a detailed description as follows:

“…Burnout was defined as a combination of a high score on emotional exhaustion and a high score on depersonalization. High scores in either of the dimensions were defined as scores exceeding the top tentile. In the current study, the Cronbach’s alpha for the MBI-HSS was 0.96. The top tertile of emotional exhaustion corresponded to a score of 27, while a score of 10 represented the threshold for depersonalization. Therefore, individual who scored >27 on the emotional exhaustion ‘and’ scored > 10 on the depersonalization met the criteria for burnout. ”

- “Further, both high-EE and high-DP (above average) were judged as the group with high-burnout”: does this mean that PA was not considered for the classification?

- Studies were cited which used a similar approach. However, the MBI provides norm-based cutoff values for "low risk," "moderate risk," and "high risk." Why were these not considered?

Response：Thank you for your reminder. We searched the literature again and examined the cutoff values provided by the MBI. But we used the revised Chinese version of the 17-item Maslach Burnout Inventory specifically tailored to the Chinese population, instead of the original 22-item version. However, there is currently no universally accepted cutoff score for this scale. We employed the top tertile as the cutoff, a method used in other study as well [16]. We added this description in the section of discussion and discussed the potential implications of tertile cutoff in our study. 

(2) Discussion, Main findings: It is useful to briefly repeat the main results, but the statistical metrics (CI, OR) should not be repeated.

Response: We have made the necessary revisions based on your suggestions. Please review it again.

(3) Discussion, Dynamic risk factors:

- I didn't entirely understand why a nighttime broadcast is less dangerous than a daytime one. Is night work the norm in this profession?

- Overall, the discussion could make more reference to theory and previous studies.

Response：We re-evaluated the data and do all of the statistical analyses again in the new dataset. We found conducting live stream during daytime remains a significant risk factor compared to evening. The public usually watch livestreams for entertainment purposes. Burnout associated with live streaming during the daytime is likely because the prime time for live streaming is usually in the evening when the public has time to watch. Based on our understanding, for high-performing, popular, and influential streamers, their work is primarily scheduled for the evening. In contrast, streamers with lower performance and influence may be assigned to stream during the daytime. This could be related to their state of burnout, but the specific mechanisms need further investigation. We add the description in the section of discussion in the revised manuscript. And also we found the simple categorization of risk factors into dynamic and static factors was not appropriate, we have abandoned this classification method in the revised paper.

(4) Discussion, Protective factor: “Interpersonal support is unlikely to directly cause burnout”: However, the analysis model tested direct influence (and it was significant), right? No moderator effects of "interpersonal support" were tested either (if I interpret it correctly). Thus, these conclusions can actually only represent hypotheses for future research.

Reponse：“interpersonal support”in the revised manuscript we used a more accurate description of “supervisor and coworker support”. Because the last 8 items of the JCQ-22 to measure the social support, instead of a professional social support scale. These 8 items consist of four items represent the support from supervisors and remaining four items measuring the support from coworkers. 

In our new multivariate logistic model, supervisor and coworker support was found to be a risk factor, rather than a protective factor, which contradicts the findings in the teacher and nurse populations. We conducted more analysis on the results and added relevant descriptions in the discussion section. Please review it again.

(5) Minor comments:

- The p-value is usually written in lower case.

- I am not a native English speaker, but a few phrases seemed to me linguistically not quite correct. I recommend having the text checked again.

Response: We have made the necessary revisions based on your suggestions. Thanks again. 

Reviewer #2: (1) How was this set of the exclusion criteria determined? The marked content is “infection of certain nervous system related diseases or mental disorders affected their communication function, for instance, general anxiety disorder, major depressive disorder

Response: we reexamined our study design and execution records. The description was not appropriate. Actually, the investigators assessed whether participants were able to read and respond to the content of the questionnaire. We revised the description as follows: Participants were excluded if they: (1) had not signed a contract with a broadcasting company; (2) were younger than 16; (3) had difficulties in reading or communicating; and (4) were unable to independently complete questionnaires using electronic devices.

(2) Provide information on the total score per dimension and the cut off for each dimension. How was personal accomplishment categorized into high or low burn out? 

Response：Through literature review, we found that the practice of simply using the total score of the Maslach Burnout Inventory-Human Services Survey (MBI-HSS) to indicate burnout does not adhere to the original scale's guidelines. Therefore, we have redefined the categorization of burnout based on the following two studies. 

[7] Rotenstein LS, Torre M, Ramos MA, Rosales RC, Guille C, Sen S, Mata DA. Prevalence of Burnout Among Physicians: A Systematic Review. JAMA. 2018;320(11):1131-1150. doi: 10.1001/jama.2018.12777. 

[16] Upton D, Mason V, Doran B, Solowiej K, Shiralkar U, Shiralkar S. The experience of burnout across different surgical specialties in the United Kingdom: a cross-sectional survey. Surgery. 2012;151(4):493-501. doi: 10.1016/j.surg.2011.09.035. 

In the revised manuscript, we provided a detailed description as follows:

“…Burnout was defined as a combination of a high score on emotional exhaustion and a high score on depersonalization. High scores in either of the dimensions were defined as scores exceeding the top tentile. In the current study, the Cronbach’s alpha for the MBI-HSS was 0.96. The top tertile of emotional exhaustion corresponded to a score of 27, while a score of 10 represented the threshold for depersonalization. Therefore, individual who scored >27 on the emotional exhaustion ‘and’ scored > 10 on the depersonalization met the criteria for burnout. ”

(3) Did the authors take 'assent' from the participants who were aged 16-18.

Response：Three participants aged 16 -18 was recruited in this study. We obtained assent from the participants themselves and consent from their guardians before the interview. 

(4) Other minor comments

Response: We have made the necessary revisions based on your suggestions. Please review it again.

---

## [Decision Letter · Decision Letter 1]

4 Mar 2024

PONE-D-23-18174R1Burnout among Chinese live streamers: prevalence and correlatesPLOS ONE

Dear Dr.<table border="0" cellpadding="0" cellspacing="0" class="datatable3" style="border-collapse: collapse; width: 678px; line-height: 14px; color: rgb(0, 0, 51); font-family: verdana, geneva, arial, helvetica, sans-serif; font-size: 11.2px;"> 

Ziwei Liu

</table>,Thank you for submitting your manuscript to PLOS ONE. After careful consideration, we feel that it has merit but does not fully meet PLOS ONE’s publication criteria as it currently stands. Therefore, we invite you to submit a revised version of the manuscript that addresses the points raised during the review process.

 Please submit your revised manuscript by Apr 18 2024 11:59PM. If you will need more time than this to complete your revisions, please reply to this message or contact the journal office at plosone@plos.org. Please include the following items when submitting your revised manuscript:A rebuttal letter that responds to each point raised by the academic editor and reviewer(s). You should upload this letter as a separate file labeled 'Response to Reviewers'.A marked-up copy of your manuscript that highlights changes made to the original version. You should upload this as a separate file labeled 'Revised Manuscript with Track Changes'.An unmarked version of your revised paper without tracked changes. You should upload this as a separate file labeled 'Manuscript'.If applicable, we recommend that you deposit your laboratory protocols in protocols.io to enhance the reproducibility of your results. Protocols.io assigns your protocol its own identifier (DOI) so that it can be cited independently in the future. For instructions see: https://journals.plos.org/plosone/s/submission-guidelines#loc-laboratory-protocols. Additionally, PLOS ONE offers an option for publishing peer-reviewed Lab Protocol articles, which describe protocols hosted on protocols.io. Read more information on sharing protocols at https://plos.org/protocols?utm_medium=editorial-email&utm_source=authorletters&utm_campaign=protocols.

We look forward to receiving your revised manuscript.

Kind regards,

Othman A. Alfuqaha, Ph.D.

Academic Editor

PLOS ONE

Journal Requirements:

Reviewers' comments:

Reviewer's Responses to Questions

**Comments to the Author**

1. If the authors have adequately addressed your comments raised in a previous round of review and you feel that this manuscript is now acceptable for publication, you may indicate that here to bypass the “Comments to the Author” section, enter your conflict of interest statement in the “Confidential to Editor” section, and submit your "Accept" recommendation.

Reviewer #1: (No Response)

2. Is the manuscript technically sound, and do the data support the conclusions?

Reviewer #1: Yes

3. Has the statistical analysis been performed appropriately and rigorously? 

Reviewer #1: Yes

4. Have the authors made all data underlying the findings in their manuscript fully available?

Reviewer #1: Yes

5. Is the manuscript presented in an intelligible fashion and written in standard English?

Reviewer #1: Yes

6. Review Comments to the Author

Reviewer #1: Reviewer 1, second review:

I would like to thank the authors for their conscientious revision of the manuscript, taking into account the reviewers' comments.

Most of the points are now clarified to me.

I only have one follow-up question concerning burnout classification:

I understand that the Chinese version has a different number of items and that there are no meaningful threshold values. The procedure with the combined risk as a criterion also seems reasonable to me.

However, the relevance of personal accomplishment is still not clear to me. Although the subscale is mentioned, it is not taken into account for the burnout classification, only emotional exhaustion and depersonalization. Yet, low personal accomplishment would also be an indicator of burnout. Is there a reason why this subscale was not included in the chosen approach?

7. PLOS authors have the option to publish the peer review history of their article (what does this mean?). If published, this will include your full peer review and any attached files.

Reviewer #1: **Yes: **Verena Hofmann

---

## [Author Response · Author response to Decision Letter 1]

7 Mar 2024

Response to Reviewers

Reviewer comments:

I would like to thank the authors for their conscientious revision of the manuscript, taking into account the reviewers' comments.

Most of the points are now clarified to me.

I only have one follow-up question concerning burnout classification:

I understand that the Chinese version has a different number of items and that there are no meaningful threshold values. The procedure with the combined risk as a criterion also seems reasonable to me.

However, the relevance of personal accomplishment is still not clear to me. Although the subscale is mentioned, it is not taken into account for the burnout classification, only emotional exhaustion and depersonalization. Yet, low personal accomplishment would also be an indicator of burnout. Is there a reason why this subscale was not included in the chosen approach?

Response: Initially, we used the following criteria to determine burnout：higher scores on the emotional exhaustion and depersonalization subscales and lower scores on the personal accomplishment subscale correspond to higher levels of burnout。[Rotenstein LS, Torre M, Ramos MA, Rosales RC, Guille C, Sen S, Mata DA. Prevalence of Burnout Among Physicians: A Systematic Review. JAMA. 2018 Sep 18;320(11):1131-1150. doi: 10.1001/jama.2018.12777. PMID: 30326495; PMCID: PMC6233645.]

However, the statistical analysis results showed that only 3 of 343 participants met the criteria: EE> 27, DP >10 ‘and’ PA<17 (the bottom tertile among the 343 participants). This clearly appears to be an unreasonable result.

Maslach in his article mentioned that personal accomplishment cannot be assumed to be the opposite of EE and DP. The correlations between the personal accomplishment subscale and the other subscales are low. [Maslach, Christina & Jackson, Susan & Leiter, Michael. (1997). The Maslach Burnout Inventory Manual.] But in our sample, we found that personal accomplishment was not negatively associated with emotional exhaustion and depersonalization but rather positively and significantly associated with them. In other words, individuals with higher levels of emotional exhaustion and depersonalization also exhibited higher levels of personal accomplishment. Despite excluding the possibility of data errors, it remains unclear whether this phenomenon is a unique characteristic of the live streamer population or if it is influenced by other confounding factors. This necessitates further investigation through an independent study to explore the underlying reasons.

 The plot is missing, please see the attachment file.

Considering the strong correlation between personal accomplishment and emotional exhaustion or depersonalization, we did not use the PA subscale to assist in determining burnout in this study. We used the criteria reported in other studies as follow, which consider only scores higher than 27 on the emotional exhaustion scale ‘and’ scores higher than 10 on the depersonalization scale to determine burnout.

[Pedersen AF, Ingeman ML, Vedsted P. Empathy, burn-out and the use of gut feeling: a cross-sectional survey of Danish general practitioners. BMJ Open. 2018 Feb 28;8(2):e020007. doi: 10.1136/bmjopen-2017-020007. PMID: 29490966; PMCID: PMC5855338.]

[Shanafelt TD, West CP, Sloan JA, Novotny PJ, Poland GA, Menaker R, Rummans TA, Dyrbye LN. Career fit and burnout among academic faculty. Arch Intern Med. 2009 May 25;169(10):990-5. doi: 10.1001/archinternmed.2009.70. PMID: 19468093.]

[Upton D, Mason V, Doran B, Solowiej K, Shiralkar U, Shiralkar S. The experience of burnout across different surgical specialties in the United Kingdom: a cross-sectional survey. Surgery. 2012 Apr;151(4):493-501. doi: 10.1016/j.surg.2011.09.035. Epub 2011 Nov 16. PMID: 22088818.]

[Shanafelt TD, Boone S, Tan L, Dyrbye LN, Sotile W, Satele D, West CP, Sloan J, Oreskovich MR. Burnout and satisfaction with work-life balance among US physicians relative to the general US population. Arch Intern Med. 2012 Oct 8;172(18):1377-85. doi: 10.1001/archinternmed.2012.3199. PMID: 22911330.]

---

## [Editor Report · Decision Letter 2]

26 Mar 2024

Burnout among Chinese live streamers: prevalence and correlates

PONE-D-23-18174R2

Dear Dr. Liu,

We’re pleased to inform you that your manuscript has been judged scientifically suitable for publication and will be formally accepted for publication once it meets all outstanding technical requirements.

Kind regards,

Jenny Wilkinson, PhD

Academic Editor

PLOS ONE

Additional Editor Comments (optional):

Thank you for your responses, these have satisfactorily addressed the reviewer comments.
---

## [Editor Report · Acceptance letter]

10 May 2024

PONE-D-23-18174R2 

PLOS ONE

Dear Dr. Liu, 

I'm pleased to inform you that your manuscript has been deemed suitable for publication in PLOS ONE. Congratulations! Your manuscript is now being handed over to our production team.

Kind regards, 

on behalf of

Dr Jenny Wilkinson 

Academic Editor

PLOS ONE